# Dissecting Melanoma Ecosystem Heterogeneity from Molecular Characteristics to Genetic Variation at Single-Cell Resolution

**DOI:** 10.3390/ijms26209956

**Published:** 2025-10-13

**Authors:** Congxue Hu, Liyuan Li, Tengyue Li, Baobin Qi, Wanqi Mi, He Yu, Kaiyue Yang, Qi Ou, Xia Li, Yunpeng Zhang

**Affiliations:** College of Bioinformatics Science and Technology, Harbin Medical University, Harbin 150081, China; hucx1996@hrbmu.edu.cn (C.H.); liliyuan9501@163.com (L.L.); travilucas@163.com (T.L.); 18853496439@163.com (B.Q.); miwanqi_9657@163.com (W.M.); 2020020465@hrbmu.edu.cn (H.Y.); yangkaiyue0904@hrbmu.edu.cn (K.Y.); ouqi_2000@163.com (Q.O.)

**Keywords:** melanoma heterogeneity, molecular characteristics, genetic variation, single cell, melanoma treatment

## Abstract

Melanoma shows heterogeneity across body sites like skin, acral skin, and the uvea, driven by molecular characteristics and genetic variations. However, comparative studies exploring the heterogeneity of melanoma across different anatomical sites remain limited, hindering a comprehensive understanding of its underlying biology. We proposed a research framework through bioinformatics to analyze the tumor ecosystems of cutaneous, acral, and uveal melanoma, from molecular characteristics to genetic variations at single-cell resolution. We found that oxidative phosphorylation (OXPHOS) is a critical driver of tumor cell evolution, with abnormal ribosomal gene and tumor suppressor expression observed in uveal melanoma (UM). Additionally, we screened for potential drug targets and drugs against tumor cells. In the immune microenvironment, acral melanoma (AM) and UM exhibit stronger immunosuppressive characteristics compared to cutaneous melanoma (CM). OXPHOS contributes to T cell cytotoxicity dysregulation in CM and AM, while interferon-γ is crucial in UM. Tumor cells may also induce T cell dysfunction through biological signals such as MIF-CD74 and HLA-E-NKG2A. This study offers valuable insights into melanoma heterogeneity, providing a comprehensive research framework for understanding the distinct molecular and immune characteristics of CM, AM, and UM, and potentially guiding the development of therapeutic strategies tailored to each melanoma subtype.

## 1. Introduction

Genomic instability and mutation have been identified as important initiators of cancer cell molecular characteristics and Hallmarks and are closely associated with patient prognosis [1,2]. Copy number variations (CNVs) in genetic variation is one of the main ways to cause genomic instability and mutation in cancer [3]. Dissecting this genetic variation is critical to understanding the heterogeneity of the melanoma ecosystem, as the molecular characteristics of melanomas at different body sites are largely driven by genetic variation [4]. This is related to ultraviolet radiation (UVR) [5]. Different parts of the body are exposed to different areas of sunlight, so the degree of genetic variation and mutation burden driven by UVR is also different [5]. Non-acral skin sites are more exposed to UVR and have a larger exposure area. Melanoma cells in these sites exhibit high mutation burdens and genetic variations driven by UVR. Acral sites are often shielded from sunlight and receive much less UVR compared to non-acral skin. Uveal melanoma (UM), located in the eye, is typically not caused by UVR [5]. Although genomic instability and mutation burden in melanoma cells of acral skin and uveal sites is lower than that of non-acral skin, whole-genome sequencing study [4] indicate that these two types of melanomas exhibit a certain degree of genetic variation. In conclusion, genetic variations contribute to the heterogeneity in the molecular characteristics and tumor ecosystems of melanomas across different body sites. However, this aspect remains underexplored and warrants further investigation to better understand its impact.

Single-cell transcriptomics (scRNA-seq) and bioinformatics provide novel perspective and research methods for dissecting this aspect, providing valuable insights into its molecular characteristics, genetic variations, and tumor ecosystem heterogeneity [6]. Presently, single-cell transcriptomics have been used to study melanomas. Zhao H and colleagues compared the differences in tumor cell subpopulations and microenvironment between cutaneous melanoma (CM) and acral melanoma (AM) [7]. Zhang C and colleagues compared the heterogeneity between cutaneous melanoma and acral melanoma in terms of tumor and immune phenotypes [8]. Wei C and colleagues analyzed the early dissemination mechanisms of AM [9]. However, these studies have largely focused on limited melanoma subtypes, and a comprehensive single-cell comparison across multiple anatomical subtypes of melanoma remains lacking. In particular, the molecular distinctions and genetic underpinnings of uveal melanoma (UM), in contrast to CM and AM, are still poorly understood.

In this study, we systematically compared cutaneous melanoma (CM), acral melanoma (AM), and uveal melanoma (UM) from their molecular features to genetic variations at single-cell resolution, aiming to gain a deeper understanding of the heterogeneity within the melanoma tumor ecosystem. This computational and integrative bioinformatics approach enables the identification of key molecular programs that may drive melanoma subtype diversity and offers insights into potential targeted therapeutic strategies. Although our study is based on computational analyses, the insights presented here lay a strong foundation for subsequent experimental validation. Notably, this research approach and framework can be adapted and extended to analyze other types of cancer, further advancing our understanding of tumor biology.

## 2. Results

### 2.1. Single-Cell Landscape of Melanoma Ecosystem in Multiple Body Sites

To elucidate the ecosystem of melanomas in the CM, AM, and UM at the single-cell resolution, we analyzed single-cell data from 16 primary tumor samples and also collected single-cell data from 24 normal tissue samples for comparison. Detailed sample information are provided in Figure 1A and Appendix A. We obtained 130,140 cells from tumor samples and 94,348 cells from normal samples (Figure 1B,C and Appendix A), and revealed the cell types in the ecosystems of different sites based on the expression of classic marker genes (Figure 1D; Appendix A) [10,11]. Finally, we built integrated single-cell landscape of melanoma ecosystem in multiple body sites (Figure 1B).

Our study reveals significant differences in the cellular composition and proportions of the ecosystem across melanoma in three types of melanomas (Figure 1C). In normal non-acral skin, basal keratinocytes and suprabasal keratinocytes constitute the predominant cell populations. In contrast, normal acral skin, in addition to containing the aforementioned cell types, also exhibits a higher abundance of fibroblasts. In the choroid, endothelial cells and fibroblasts are the major cell types, and this body site also contains vision-related cells such as photoreceptor cells, retinal pigment epithelium cells, and muller glia cells. In CM, AM, and UM, melanoma cells are the principal cellular constituents. Compared to normal tissues, there is a general increase in the proportion of immune cells within tumor, including T cells, B cells, and monocytes/macrophages. These results indicate that cell types exhibit site-specific characteristics. Additionally, the cellular composition of the three types of melanomas has undergone significant changes compared to normal tissues.

Despite the fact that the proportions of cell types may be influenced by a variety of biological and environmental factors, the undeniable significant differences in cell types and their relative abundances across different body sites are a notable characteristic [5,6]. These differences impact the biological characteristics of melanoma. We identified specific cell types that differ in abundance at single-cell resolution.

### 2.2. Site-Specific Molecular Characteristics and Biological Functions in Melanoma Tumor Cells

To gain an in-depth understanding of the molecular characteristics of melanoma cells from different body sites, we performed clustering analysis on normal melanocytes and melanoma tumor cells (Figure 2A,B), ensuring that batch effects were minimized during the clustering process. Before batch correction, the average iLISI score was 1.148, indicating a notable batch effect. After batch correction, the average iLISI increased to 1.863, reflecting a significant reduction in batch effects (Appendix A). The low level of residual batch effects suggests minimal interference with the biological signals, providing a robust foundation for analyzing the heterogeneity of melanoma tumor cells.

Figure 2A clearly differentiates the distribution between normal melanocytes and melanoma tumor cells. Further cellular classification revealed that in Figure 2B, normal melanocytes exhibit a propensity to aggregate into distinct clusters. The distribution of melanoma tumor cells exhibits significant site-specific heterogeneity. Specifically, CM and AM share certain cellular clusters, as identified in the subgroup division (clusters 2, 8, 12, and 13), while AM display unique clusters (cluster 6). In contrast, the clustering characteristics of UM are significantly different from those of CM and AM, with the exception of cluster 12 being shared with CM and AM, all other clusters within UM are unique (cluster 0, 1, 3, 4, 5, 9, 10, and 11). These indicate that at single-cell resolution, tumor cells in the three types of melanomas ecosystems exhibit some common molecular characteristics while also having unique molecular features.

Molecular characteristics of melanocytes and melanoma cells in three body sites suggest that their biological functions exhibit body site-specificity (Figure 2C; Appendix A). Melanocytes in non-acral skin are predominantly involved in developmental pigmentation. Melanocytes in acral skin engage in skin morphogenesis. Melanocytes in choroid are associated with UV damage repair and camera-type eye. Additionally, melanocytes in these three sites also exhibit consistent biological functions, such as melanocyte differentiation, pyruvate transmembrane transport, and maintenance of adherens junctions. We have also compared the differences in biological functions between melanoma tumor cells and melanocytes from three body sites (Appendix A). Normal melanocytes are generally associated with cell differentiation and morphogenesis. CM tumor cells demonstrate characteristics of integrin binding, focal adhesion, and epithelial–mesenchymal transition. The characteristics of AM and UM tumor cells are mainly related to OXPHOS and ATP metabolic process. These results indicate that the biological functional characteristics of melanocytes and melanoma cells in three body sites reveal site-specificity.

Additionally, we utilized a melanoma feature gene set in Appendix A to score melanoma tumor cells and observed that the scores for differentiation and neural crest-like states were low (Figure 2D and Appendix A), which is consistent with previous reports [5]. Moreover, by assessing cellular activity-related features such as proliferation, DNA repair, DNA damage, and cell cycle, the tumor cells could be categorized into active and inactive states (Figure 2D).

Notably, CM and AM are predominantly composed of active cells. In contrast, UM not only contains active cells but also incorporates inactive cells that exhibit significant differential expression of ribosomal protein genes, including RPL19, RPL26, RPS13, etc. (Figure 2E; Appendix A). These indicate that ribosomal genes in UM tumor cells are abnormally expressed, which may affect the cell cycle-related activity characteristics of melanoma cells. Strikingly, the expression of TP53, PTEN, and RB1, tumor suppressors activated in response to ribosomal stress, was significantly upregulated in inactive cells compared to active cells (Appendix A). These observations raise the possibility that aberrant ribosome biogenesis may trigger tumor suppressor-mediated responses [12,13,14], thereby contributing to the maintenance of non-proliferative, inactive cell state in a subset of UM cells. Moreover, to exclude the possibility that the presence of inactive cells was due to inter-sample heterogeneity, we quantified the proportion of inactive cells in each UM sample. The consistent presence of inactive cells across multiple samples suggests that this subpopulation reflects a genuine biological feature rather than technical or sampling variation (Appendix A).

In general, while melanocytes and melanoma tumor cells in three body sites exhibit some shared characteristics, their molecular characteristics and biological functions also reveal significant site-specificity. Notably, we have identified a population of inactive tumor cells expressing ribosomal genes in UM, a finding that may point to the distinct molecular characterization of UM.

### 2.3. Deciphering Genetic Variation from Cell to Molecular Levels in the Ecosystem of Three Types Melanomas

In Section 2.2, we conducted a re-clustering analysis of tumor cells. After removal batch effects, the molecular characteristics and biological functions of tumor cells exhibited highly body site-specific patterns at the biological level. Genetic variation is an important driver of changes in cellular molecular characteristics [3,15]. We hypothesize that these molecular characteristics within melanoma ecosystems may be driven by genetic variations among tumor cells. Therefore, we verified this by examining copy number variations (CNVs) from single-cell, chromosomal arm, and molecular perspective, and we have confirmed this hypothesis using InferCNV analysis.

The results at the single-cell level revealed significant body-site heterogeneity in the CNV patterns across different sites and samples, as well as a certain degree of intra-tumoral heterogeneity (Appendix A). Using normal melanocytes as a control, we categorized tumor cells into three degrees based on CNV scores: low, medium, and high (Figure 3A and Appendix A). Figure 3A also illustrates that melanoma tumor cells generally exhibit higher CNV scores than normal cells, indicating a significant increase in genomic instability within the tumor cells. Notably, there were also differences in the extent of CNVs among tumor cells from the three body-site ecosystems (Figure 3A): CM and AM tumor cells predominantly showed medium to high levels of CNV, while UM was characterized by a predominance of tumor cells with low CNV levels; this phenomenon we found at single-cell resolution is consistent with previous reports from comparative genomics studies [4]. We also observed a close correlation between the levels of CNV and the functional features of tumor cells. As depicted in Figure 3A,B, active tumor cells exhibited higher CNV levels compared to inactive cells, suggesting that CNV events may influence critical biological processes such as tumor cell proliferation, DNA repair, DNA damage, and cell cycle. Additionally, we observed that in tumor cells with low CNV levels, particularly in UM, melanoma tumor characteristics were not pronounced, the scores for melanoma features increased with rising CNV scores (Figure 3C and Appendix A). These phenomena hold significant implications for understanding the pathogenesis and heterogeneity of melanoma. By deciphering single-cell CNV patterns across the three types of melanomas, we have elucidated the pivotal role of CNVs in the malignant transformation and tumorigenesis of melanoma cells.

Using normal melanocytes as a reference, we systematically delineated the chromosomal CNV landscape across melanoma subtypes and identified both canonical and previously underreported CNV events (Figure 4A and Appendix A). Several high-scoring CNVs were consistent with publicly available datasets such as the Broad Institute Fire Browse portal (http://firebrowse.org/) and previous studies [4]. Specifically, we observed gains of 7p, 15q, 17q, and 22q, and LOH of 2q, 6p, 6q, 8p, 9p, and 11q in CM. In AM, we noted gains of 1q, 6p, 11q, 12q, 20q, and 22q, and LOH of 6q and 11q. UM exhibited gains of 8q and 22q, and LOH of 3p and 3q. Additionally, we detected CNVs that have been less characterized or less studied, including gains of 4p and LOH of 19q in CM, gain of 5q in AM, and gains of 12q and 15q in UM. Importantly, our analysis also revealed CNV alterations that are rarely documented in previous melanoma studies, including gains of 4p and LOH of 19q in CM, gain of 5q in AM, and gains of 12q and 15q in UM. These findings reveal tumor type-specific CNV events that may have been previously overlooked and deserve further exploration.

We also revealed a comprehensive landscape of CNVs by identifying genes that exhibited both CNVs and significant differential expression (DEG with CNV) in tumor cells (Figure 4B and Appendix A), using normal melanocytes as a reference. These genes showed both commonalities and differences in the three types of melanomas (Figure 4B and Appendix A). Further analysis revealed the impact of these genes on the melanoma ecosystem, which varied significantly in three sites (Figure 4C; Appendix A). In CM, biological functions such as epidermis development, humoral immune response, and regulation of inflammatory response were suppressed, while integrin binding and focal adhesion were activated; In AM, positive regulation of cell death, UV response Dn, and TNFA signaling via NFKB were suppressed, while oxidative phosphorylation, ATP biosynthetic process, and negative regulation of cell killing were activated; in UM, eye morphogenesis, cytoskeleton organization, and G2M checkpoint were suppressed, and ribosome assembly, oxidative phosphorylation, and interferon response were activated (Figure 4C,D). These findings are consistent with the observations from Section 2.2, indicating that CNVs associated with genetic variations in tumor cell are key factors driving the molecular characteristics, biological functions, and heterogeneity of the melanoma ecosystem.

Beyond conventional arm-level CNV profiling, we performed high-resolution CNV mapping at the molecular level, uncovering subtle yet biologically relevant alterations and revealing a previously underrecognized level of genomic complexity and anatomical-site-specific heterogeneity among melanoma subtypes. These results highlight the added value of integrating single-cell resolution with normal-cell references to refine CNV characterization, offering new perspectives on the genomic evolution of melanoma beyond what has been described in previous studies.

### 2.4. The Evolutionary Processes Generate Diverse Malignant Transcriptional Programs in Three Types of Melanomas

Genetic variations are closely linked to the evolution of tumor cells [1,3,15], and our goal is to identify the differentially expressed genes with CNV event (DEG with CNV) that drive the malignant progression of melanoma tumor cells.

We used monocle3 to construct the evolutionary trajectories of melanoma tumor cells from three body sites, revealing complex evolutionary paths that melanoma tumor cells undergo as they transition from normal melanocytes through various routes. To assess potential batch effects in trajectory inference, we computed the average iLISI score, which was 1.467, indicating minimal batch interference (Appendix A). This low level of batch effect suggests that the inferred trajectories are robust and predominantly driven by underlying biological differences, supporting their validity in delineating the malignant evolution of melanoma cells.

Along the evolutionary trajectories of melanoma tumor cells, their state progressively shifts from inactive to active, accompanied by an increase in CNV levels (Figure 5A), and the emergence of diverse malignant transcriptional programs (Figure 5B–J). In CM and AM, the evolutionary trajectory programs involve the transformation of normal melanocytes into active tumor cells (program1, program3), and a smaller subset into inactive tumor cells (program2, program4) (Figure 5B,E). In UM, there are evolutionary trajectories from normal melanocytes to active tumor cells (program5), and from inactive to active tumor cells (program6) (Figure 5H).

We further identified transcriptomic gene modules that changed within different trajectory programs (Figure 5C,F,I) and characterized the malignant transcriptional programs of melanomas from the three sites based on their distribution along the trajectories (Figure 5D,G,J; Appendix A). The results show that the main biological features of program1, program3, and program5 are oxidative phosphorylation and cellular respiration, while program2 and program4 are associated with epidermis development, epidermal cell differentiation, and intermediate filament cytoskeleton organization.

Notably, program6 in UM exhibits functional characteristics related to ribosome, rRNA processing, and ncRNA processing, highlighting the central role of ribosome-associated abnormalities in the malignant progression of UM. These findings suggest the driving role of ribosome-related functional abnormalities in UM malignant progression. Intriguingly, analysis of the pseudotime trajectory further revealed that the expression of critical tumor suppressor genes—TP53, PTEN, and RB1—gradually decreases along the progression path in UM tumor cells (Appendix A). This dynamic downregulation may reflect a weakening of tumor suppressor-mediated surveillance mechanisms, potentially facilitating unchecked proliferation and malignant advancement. Additionally, OXPHOS and cellular respiration were identified as major driving factors across all three melanoma subtypes, underscoring their fundamental roles in melanoma biology.

Combined with previous results from this study, the biological functions driving melanoma malignant progression closely align with those mediated by DEG with CNV. This consistency suggests that CNVs may play a critical role in regulating key genes that contribute to the malignant transformation of melanoma, highlighting their potential as targets for therapeutic intervention. Therefore, we explore this aspect in more depth in the next section.

### 2.5. Drug Screening Based on the Malignant Transcriptional Regulatory Networks

To better explore the role of genetic variation in driving the malignant evolution of melanoma cells and its potential as a therapeutic target, we first performed Kaplan–Meier survival analysis on the malignant transcriptional programs identified in Section 2.4. Our analysis revealed a significant association between these transcriptional programs and the survival prognosis of melanoma patients.

Notably, the activation of these malignant transcriptional programs was linked to a decrease in overall survival outcomes (Appendix A). We then screened for genes within these malignant transcriptional programs that were significantly associated with survival (Appendix A) and used pySCENIC to predict the transcription factors (TFs) regulating these genes. By integrating protein interaction information from the STRING database, we constructed a regulatory network for the malignant transcriptional programs associated with prognosis (Figure 6A,C,E and Appendix A).

The results showed that the activation of regulatory networks in melanoma from three sites significantly reduced the overall survival outcomes of patients. Moreover, more than half of the nodes within these networks exhibited CNVs (Appendix A). These networks exhibited characteristics related to oxidative stress (Appendix A), such as response to oxidative stress, chemical carcinogenesis-reactive oxygen species, and integrated stress response signaling. Additionally, OXPHOS emerged as a shared feature of molecular networks across all three melanoma subtypes, whereas ribosomal pathways represented a subtype-specific hallmark of the UM network (Appendix A).

Notably, the biological characteristics of these networks closely resemble the biological functions mediated by DEG with CNV, as identified earlier in this study. These findings underscore the potential role of CNVs in driving the malignant progression of melanoma cells by influencing key biological pathways such as OXPHOS, oxidative stress, and ribosomal function. These factors are likely to contribute to the poor prognosis observed in melanoma patients.

These suggests that targeted therapies aimed at these biological features may represent new strategies for treating melanoma. The genes and TFs within these networks provide potential targets for the development of new melanoma treatments. Therefore, we used Drug2cell to screen potential drug targets and drugs within the network. The results showed that crizotinib and palbociclib can target MET in CM (Figure 6A), while ixazomib, ixazomib citrate, carfilzomib, and bortezomib can target PSMD8, PSMB6, and PSMD4 in AM (Figure 6C). Ataluren can target RPL17 in UM (Figure 6E).

The expression of these drug targets is closely associated with tumor progression and patient survival. Many target genes have undergone CNV events (Figure 6A,C,E; Appendix A), including loss of MET, PSMB6, and PSMD8, as well as gain of PSMD8. The screened drugs have stable binding capabilities with their corresponding targets (docking score < −5 kcal/mol) (Figure 6B,D,F). Notably, several of the identified drugs are FDA-approved and currently in use for cancer therapy (Appendix A). For example, Carfilzomib, a proteasome inhibitor, is already approved for treating melanoma, underscoring the therapeutic validity of our screening pipeline. This finding provides a compelling rationale for the potential repurposing of other screened drugs—such as Crizotinib, Palbociclib, and Ixazomib—which target key molecules associated with melanoma progression.

In summary, we have elucidated the significant role and heterogeneity of CNVs in driving melanoma tumor cell progression. We also identified the molecular and functional characteristics of the regulatory networks associated with three melanoma malignant transcriptional programs. Based on these findings, we systematically screened potential drug targets and therapeutic candidates for the treatment of these melanoma subtypes. The findings presented here provide a valuable resource and rational basis for future experimental investigations, and may guide translational efforts to evaluate the efficacy and therapeutic relevance of these candidate drugs in melanoma treatment.

### 2.6. Uncovering Dysregulated Signaling of Cytotoxicity in Antitumor T Cell in Three Types of Melanoma Ecosystems

We analyzed the heterogeneity of melanoma, spanning molecular characteristics to genetic variations, which may contribute to differences in treatment responses. Immunotherapy has emerged as one of the primary approaches for treating tumors, with reversing the cytotoxic dysfunction of antitumor T cells recognized as an effective antitumor immunotherapy strategy [16,17,18,19]. Therefore, in this section, we explore the heterogeneity of three melanoma subtypes by examining dysregulated cytotoxic signaling in antitumor T cells. This analysis provides insights into dysregulated immunotoxicity, offering potential strategies for immunotherapy tailored to different melanoma ecosystems.

We reclassified T cells in the three melanoma ecosystems into seven clusters, each significantly expressing different markers (Figure 7A and Appendix A), and ensured that batch effects were minimized during the clustering process. The low level of residual batch effects suggests minimal interference with the biological signals (average iLISI = 2.204) (Appendix A), providing a robust foundation for analyzing the heterogeneity of T cells. Most T cell clusters in AM and UM were distributed together, while those in CM were distributed in different regions. This distribution pattern indicates distinct antitumor immune environments among the three melanomas. Using known T cell functional markers, we identified the states of the T cell clusters (Appendix A). In CM, T cells in the “Active and proliferating” and “Active not proliferating” states were more prevalent, indicating a stronger antitumor immune response in CM. In contrast, most T cells in AM and UM were in the “Cytotoxic with exhaustion traits” state (Figure 7A). Functional scoring of different T cell clusters also supported this observation (Appendix A). These results suggest that the immune ecosystems in AM and UM are more suppressive compared to CM. Previous studies have also reported the distinct immune microenvironment characteristics of the three melanomas [8,9,20,21]. For the first time, we systematically revealed the differences in the immune ecosystems of the three melanomas at single-cell resolution.

Lineage differentiation of T cells is a critical aspect in studying T cell cytotoxic dysfunction [17]. We delineated the lineage differentiation trajectories of T cells in three types of melanomas to investigate the key biological signals involved in cytotoxic dysfunction during T cell state transitions. Starting from Naïve T cells, T cells in three types of melanoma ecosystems exhibited unique lineage differentiation paths (Figure 7B and Appendix A). To ensure that the observed trajectories reflect genuine biological processes rather than technical artifacts, we further computed the average iLISI score, which was 2.288 (Appendix A), indicating that batch-derived variation was limited in the inferred trajectories. These results strengthen the reliability of our trajectory analysis and confirm that the T cell state transitions are predominantly shaped by intrinsic biological cues.

The major biological signals in T cells at the “Cytotoxic with exhaustion traits” state varied among the three melanomas. In the CM and AM ecosystems, the predominant signals were related to OXPHOS, ATP biosynthetic processes, and cellular respiration, whereas in UM, the main signals were interferon-gamma production and leukocyte cell–cell adhesion (Figure 7B, Appendix A). We revealed the biological signals of cytotoxic dysfunction during T cell lineage differentiation in the three types of melanoma ecosystems.

We also identified the communication signals in the three types of melanoma ecosystems that might lead to cytotoxic dysfunction of antitumor T cells. T cells in the “active not proliferating” and “active and proliferating” states were defined as CTL (cytotoxic T lymphocyte), and those in the “Cytotoxic with exhaustion traits” state were defined as CTL_ex (cytotoxic with exhaustion traits T lymphocyte). By comparing the communication signals between CTL and CTL_ex with tumor cells (Figure 7C, Appendix A), we identified the following key signals affecting T cell cytotoxicity in different melanoma ecosystems: IFNG–(IFNGR1 + IFNGR2), MIF–(CD74 + CXCR4), APP–CD74, HLA–E–CD94, HLA–E–CD94 in CM; CD99–CD99, HLA–E–KLRC1, HLA–E–CD94, ADGRE5–CD55 in AM; and MIF–CD74, APP–CD74, SIRPG–CD47 in UM. Among these communication signals (Figure 7C,D), the expression of T cell response signals showed significant differences between CTL and CTL_ex, with marked changes over pseudotime. Furthermore, these biological signals were significantly associated with the survival prognosis of melanoma patients, suggesting that these communication signals may represent potential factors by which tumor cells influence T cell cytotoxic dysfunction in the three types of melanoma ecosystems. These potential communication signals offer a valuable resource for future studies aiming to dissect the molecular mechanisms of T cell dysfunction and to evaluate their possible therapeutic relevance.

## 3. Discussion

Melanoma is a highly heterogeneous malignancy and has the highest mutation rate of all cancers [22,23]. This heterogeneity is particularly pronounced across different melanoma subtypes, exhibiting body-site heterogeneity and significant differences in molecular characteristics [5,24,25]. Genetic variation plays a crucial role in driving this heterogeneity and shaping the distinct molecular characteristics of melanoma [4,5]. Therefore, we conducted a comprehensive and systematic comparison of the tumor ecosystems of cutaneous melanoma (CM), acral melanoma (AM), and uveal melanoma (UM) at single-cell resolution, from molecular characteristics to genetic variations. Based on this analysis, we explored potential therapeutic strategies tailored to each melanoma subtype. In addition, we also provide a research framework for studying cancer heterogeneity from molecular features to genetic variations, which can be used to study other cancer types.

Our research results indicate that the molecular and biological features of normal melanocytes and melanoma cells are closely related to the body site, which may be influenced by ultraviolet radiation (UVR). For normal melanocytes, we observed that melanocytes in non-acral skin sites exhibited developmental pigmentation characteristics. This suggests that melanocytes in these sites, which are more exposed to intense UVR, may adapt by exhibiting pigmentation as a protective response. In contrast, melanocytes in acral skin sites, located in UV-shielded areas, are less affected by UVR and instead display characteristics associated with skin morphogenesis. The uveal region is more sensitive, and melanocytes in this site show UV damage–repair characteristics, which may explain why UM is less frequently associated with UVR exposure [5]. Regarding melanoma tumors, CM tumor cells exhibit characteristics [5,24,25] such as integrin binding, focal adhesion, and epithelial–mesenchymal transition, which are linked to melanoma cell invasion and metastasis [26,27]. Tumor cells from AM and UM show prominent characteristics related to oxidative phosphorylation (OXPHOS) and ATP metabolic processes. As the primary pathway for ATP production and metabolic reprogramming in melanoma cells, OXPHOS is associated with melanoma growth, metastasis, and the inactivation of immune responses to therapy [28,29,30,31].

Interestingly, we identified a subgroup of inactive tumor cells in UM that exhibit high expression of ribosomal genes, yet lacked enrichment of cell activity-related programs such as cell proliferation and cell cycle progression. This seemingly paradoxical observation could potentially be attributable to the upregulation of tumor suppressor genes, including TP53, PTEN, and RB1, within this subpopulation. The elevated ribosomal gene expression may reflect aberrant ribosome biogenesis or an enhanced biosynthetic state that triggers tumor suppressor-mediated surveillance pathways [12,13,14], which in turn might inhibit cell cycle progression, effectively decoupling ribosome production from cell proliferation. These observations raise the hypothesis that the canonical coupling between ribosome biogenesis and cell cycle progression (i.e., increased ribosome production typically accompanies and promotes cell proliferation) can be selectively disrupted in specific tumor environment, revealing a non-canonical regulatory state in UM (characterized by active ribosomal gene expression despite suppression of proliferative signaling), highlighting a potentially unique regulatory mechanism in UM that warrants further experimental investigation. The presence of this ribosome-high yet proliferation-low subpopulation has important implications for therapeutic targeting. The expression of key tumor suppressors (TP53, PTEN, and RB1) within these cells suggests the existence of a tumor-suppressive programs that could potentially be reactivated or reinforced through pharmacological intervention. This finding reveals a potentially targetable vulnerability in UM and supports therapeutic strategies aimed at harnessing intrinsic tumor-suppressive mechanisms.

Genetic variation is a key driver of tumor formation, evolution, progression, and heterogeneity [15,32,33]. Our study revealed that copy number variations (CNVs) in melanoma tumor cells are a primary determinant of their molecular and biological characteristics, and are also an important factor in the heterogeneity of different subtypes of melanoma ecosystems.

At the single-cell level, we found that the degree of CNV in UM was significantly lower than in CM and AM. The degree of CNV was strongly associated with key oncogenic processes, including tumor cell proliferation, cell cycle regulation, DNA repair, and damage response, further underscoring its functional relevance.

At the chromosomal arm level, several aneuploidy patterns we observed—particularly those affecting specific chromosomal arms—are consistent with previous reports and known to play roles in melanoma initiation and progression [34,35,36,37,38]. Notably, we also identified several previously underreported and subtype-specific CNV events that may contribute to the molecular heterogeneity of melanoma. For instance, in CM, we observed recurrent gain of chromosome arm 4p and loss of heterozygosity (LOH) on 19q—alterations that are rarely described in prior melanoma genomic studies. In AM, gain of 5q emerged as a distinct and potentially subtype-restricted aberration. Furthermore, in UM, we identified recurrent gains on 12q and 15q, which have not been extensively characterized in the context of this subtype. These findings not only expand the known CNV landscape of melanoma but also highlight the potential existence of tumor type-specific genomic vulnerabilities that warrant further investigation. Moreover, we observed that CNV-associated differentially expressed genes on affected chromosome arms were closely linked to tumor cell states, including proliferation and immune-related programs, emphasizing that genetic variation is not merely structural, but functionally consequential at the transcriptomic level. Overall, these findings not only shed light on tumor type-specific CNV alterations, but also complement existing knowledge and suggest the presence of underrecognized genomic events in melanoma that warrant further exploration.

We also found that genetic variation in melanoma cells plays a critical role in driving the distinct malignant transcriptional programs and malignant evolution of the three melanoma subtypes. Notably, more than half of the genes in the “prognosis-related malignant transcriptional regulatory network” of melanoma cells exhibit CNVs, suggesting that CNVs may be important factors in poor patient prognosis. Key features of these networks, including oxidative stress and OXPHOS, are central mechanisms promoting the malignant progression of melanoma cells. Electron leakage during the OXPHOS process increases reactive oxygen species (ROS) production [31,39,40,41], and excessive ROS levels due to an imbalance are a significant driver of melanoma development and progression [42,43]. Research has shown that melanocytes are particularly sensitive to oxidative stress, especially those induced by ROS [42]. Our findings extend previous knowledge by uncovering a potential mechanistic link between OXPHOS dysregulation and elevated ROS levels that may drive the malignant evolution of melanoma cells. While earlier studies have established the general role of ROS in melanocyte stress responses and tumorigenesis, our single-cell analysis highlights OXPHOS may be a central upstream regulator of ROS accumulation specifically within malignant melanoma subtypes, providing a more refined understanding of the molecular underpinnings of melanoma progression and highlights potential therapeutic targets.

In addition, we further uncovered a ribosome-related malignant transcriptional program that may be linked to tumor suppressor-mediated surveillance mechanisms during UM progression. Specifically, we observed that in inactive tumor cell subpopulations with high ribosomal gene expression in UM, key tumor suppressor genes including TP53, PTEN, and RB1 were concurrently upregulated [12,13,14]. This raises the hypothesis that excessive ribosome biogenesis could induce ribosomal stress, thereby activating growth-suppressive surveillance pathways that limit cell proliferation. However, pseudotime analysis revealed a gradual decline in the expression of these tumor suppressor genes along the inferred trajectory, coinciding with attenuation of the surveillance response. This might contribute to the hyperproliferation of UM tumor cells and thus the transition to an active state. Together, these observations are consistent with the hypothesis of a dynamic interplay between ribosome biogenesis and tumor suppressor-mediated control in UM cells. They point to a previously underappreciated regulatory axis that may govern the switch from inactive state to active state in UM malignant progression. Our study not only highlights the potential mechanistic novelty of this ribosome–tumor suppressor interaction in UM but also suggests it as a hypothesis for therapeutic targeting and a focus for future investigations.

Given the association of the regulatory network with poor patient prognosis, we screened potential therapeutic drugs and their corresponding targets based on these networks, revealing that many of the genes encoding these targets exhibited CNV. This finding highlights the potential prognostic and therapeutic significance of CNVs in melanoma patients. Molecular docking analyses confirmed the stability and binding patterns of these drugs to their targets. Many of these drugs have previously been investigated for melanoma treatment. For instance, Crizotinib combined with afatinib has been shown to induce cytotoxicity and mediate the death of CM tumor cells [44]. Palbociclib is also considered a potential drug for melanoma treatment [45]. Ixazomib combined with IFN-α has been used to treat advanced melanoma [46]. These results confirm the therapeutic potential of our identified drugs, and some less-explored drugs may also offer promising options for melanoma treatment.

In addition to tracing melanoma tumor cell heterogeneity from molecular characteristics to genetic variations, we also focused on the study of immunotherapy for the three melanoma subtypes. Restoration of T cell cytotoxicity is a key strategy for immunotherapy [16,17,18,19]. However, the efficacy of conventional immunotherapy is often limited in AM and UM due to the complexity and high heterogeneity of melanoma [47,48,49]. For the first time, we performed a comprehensive comparison of the immune microenvironments within the tumor ecosystems of CM, AM, and UM—consistent with prior observations [8,9,20,21]—our study uniquely explore the subtype-specific mechanisms underlying T cell dysfunction. By reconstructing the lineage differentiation trajectories of T cell states, we identified distinct regulatory programs driving cytotoxic dysfunction in each melanoma subtype, thereby establishing a direct mechanistic link between distinct molecular programs and T cell functional impairment.

Specifically, we found that oxidative phosphorylation (OXPHOS) acts as a critical driver of T cell dysfunction in CM and AM, corroborating recent findings of a subset of CD8^+^ T cells characterized by high OXPHOS activity and impaired cytotoxicity within the tumor milieu [50,51]. Notably, we further refined this observation by revealing that “cytotoxic with exhaustion traits” T cell state occupies a putative intermediate transitional state—between effective cytotoxicity and full exhaustion—suggesting potential plasticity and therapeutic reversibility. Our analysis revealed that in UM, dysregulated cytotoxicity may be primarily driven by heightened interferon-gamma (IFN-γ) signaling. Although IFN-γ plays a well-established role in enhancing antitumor immunity, it can paradoxically promote immune escape [52,53]. This has been particularly associated with poor responses to immune checkpoint blockade therapy in melanoma patients [54,55]. Consistently, our findings reveal a “cytotoxic with exhaustion traits” T cell state that may represent a putative intermediate state in UM, suggesting a potential role for aberrant IFN-γ signaling in shaping the immunosuppressive microenvironment in this subtype. Regarding ‘cytotoxic with exhaustion traits’ T cells, a previous study also reported the discovery of a possible transitional cell subset between effector memory T cells and exhausted-like T cells in melanoma [56], further supporting the potential existence of this intermediate transitional state. We further analyzed the characteristics and mechanisms of ‘cytotoxic with exhaustion traits’ T cell across the three melanoma subtypes. We believe that future studies could further verify through experiments that ‘cytotoxic with exhaustion traits’ represent an intermediate transitional state of T cells, as well as the biological functional characteristics they exhibit. Together, these results not only underscore the subtype-specific heterogeneity of T cell functional transitions but also offer novel insights into the underlying molecular programs that govern these changes. By identifying previously unappreciated metabolic and cytokine-driven pathways contributing to T cell dysfunction, our study provides a conceptual and mechanistic framework for the development of tailored immunotherapeutic strategies across melanoma subtypes.

We further investigated intercellular communication between antitumor T cells and melanoma tumor cells, revealing candidate signaling interactions that may contribute to impaired T cell cytotoxicity. Notably, the MIF–CD74 axis was prominently activated in both CM and UM. While previous studies have reported that MIF–CD74 signaling promotes melanoma cell survival and correlates with poor overall patient prognosis [57,58], its role in directly mediating T cell dysfunction has been insufficiently addressed. Our findings raise the possibility that tumor-derived MIF could engage CD74 on T cells, thereby disrupting their cytotoxic function and facilitating immune escape. Similarly, we observed that the HLA-E–NKG2A interaction was enriched in CM and AM, consistent with reports that this pathway contributes to immune evasion [59,60], and that blockade of NKG2A can restore CD8^+^ T cell activity [60]. While the precise mechanisms by which these signaling pathways influence immunity remain to be fully elucidated, our observations provide new hypotheses for how tumor cells may communicate with antitumor T cells through MIF–CD74 and HLA-E–NKG2A, potentially contributing to cytotoxic dysfunction. Beyond these previously reported axes, we also uncovered several underexplored communication signals that may shape the immune landscape of the melanoma microenvironment. Collectively, our findings deepen the understanding of context-specific mechanisms that could drive T cell dysfunction and immune evasion across melanoma subtypes, and highlight candidate immunoregulatory pathways that merit further investigation as possible therapeutic avenues. These insights lay a foundation for future experimental and translational studies aimed at clarifying their immunotherapeutic potential in CM, AM, and UM.

We performed strict quality control on the single-cell transcriptomic data (10X Genomics Chromium platform), applied Harmony to remove batch effects, and used iLISI to assess batch effects. We also performed inference of CNVs from scRNA-seq data. While potential biases may exist, many of our findings have been corroborated by previous studies. For instance, the clustering results of our cells are consistent with those observed in previous melanoma studies [7,9,61], several CNV events observed in melanoma subtypes are consistent with those reported in the Broad Institute Fire Browse portal (http://firebrowse.org/) and previous whole-genome sequencing studies [4]. Furthermore, future studies can build on this research to gain a more comprehensive understanding of melanoma biology through experimental validation and explore new therapeutic opportunities, including the validation of the drug candidates we identified.

## 4. Materials and Methods

### 4.1. Sample Information

Samples and data used in this study were obtained from the Gene Expression Omnibus (GEO) [62] database (https://www.ncbi.nlm.nih.gov/geo/, accessed on 14 March 2023). The accession number for non-acral skin samples is GSE193304, for acral skin samples are GSE179162 and GSE202352, for choroid samples is GSE135922, for cutaneous melanoma samples is GSE215120, for acral melanoma samples is GSE189889, and for uveal melanoma samples is GSE139829. Detailed sample accession information is provided in Appendix A. Data used for survival analysis were obtained from The Cancer Genome Atlas (TCGA) database (https://www.cancer.gov/tcga, accessed on 1 September 2023).

### 4.2. Quality Control of Single-Cell Sequencing Data

Single-cell data were processed using the basic pipeline of the R package Seurat [63] (version 4.4.0). After quality control, cells were filtered out if they had more than 300 unique molecular identifiers (UMIs), fewer than 200 genes expressed, more than 7500 genes expressed, and mitochondrial content greater than 20%. A total of 224,488 remaining cells were selected for subsequent analysis.

### 4.3. Dimensionality Reduction, Clustering and Annotation of Single-Cell

Dimensionality reduction and clustering of single-cell data were performed using the standard workflow of the Seurat package (version 4.4.0). Initially, the “NormalizeData” function (default parameters, version 4.4.0) was used to normalize the gene expression matrix of cells, followed by scaling and centering the data using the “ScaleData” function (default parameters, version 4.4.0). Principal components (PCs) were then computed using the “RunPCA” function (default parameters, version 4.4.0) with 3000 variable genes, and calculate the significant PCs (PCs cumulative contribution greater than 90%, individual PC contribution to variance less than 5%, or difference between consecutive PCs less than 0.1%). Clustering with the “FindNeighbors” (dims = significant PCs) and “FindClusters” functions (resolution = c(1:10/10), version 4.4.0). The “clustree” function (version 0.5.1) was used to select the appropriate resolution from the clustree R package (version 0.5.1) [Clustering trees: a visualization for evaluating clustering at multiple resolutions]. Dimensionality reduction with the “RunTSNE” function (default parameters, version 4.4.0). Specific marker genes for each cell cluster were identified using the “FindAllMarkers” (min.pct = 0.25, logfc.threshold = 0.5, version 4.4.0) function in Seurat. Subsequently, cell types for each cluster were annotated using marker genes collected from the CellMarker 2.0 database [10] (http://bio-bigdata.hrbmu.edu.cn/CellMarker/, accessed on 12 April 2023) and The Human Protein Atlas database [11] (https://www.proteinatlas.org/, accessed on 12 April 2023). The marker genes used for annotation are listed in Appendix A.

### 4.4. Batch Effect Validation and Correction

To mitigate potential technical variations introduced by integrating datasets from different sources, we evaluated and corrected batch effects across the scRNA-seq data encompassing three melanoma subtypes. Batch effects were quantitatively assessed using the integration Local Inverse Simpson’s Index (iLISI) [64], which effectively measures the extent of technical variation across batches in scRNA-seq data. An iLISI score of 1 indicates a strong batch effect, while higher scores reflect better mixing and reduced batch-associated bias.

We first applied the lisi [64] R package (version 1.0) to assess batch effects among samples from different sources within each melanoma subtype. Dimensionality reduction and clustering analyses revealed notable batch-associated separation among melanoma cells, with an average iLISI score of 1.148, suggesting significant batch effects that could confound downstream biological interpretations. To correct for these effects, we applied the “RunHarmony” function (lambda = 1, sigma = 1, max_iter = 20, nclust = 1, early_stop = TRUE, version 1.2.0) from the Harmony [64] R package (version 1.2.0). After batch correction, we reprocessed the data for dimensionality reduction and clustering. Post-Harmony integration showed a substantial reduction in batch effects, with the average iLISI increasing to 1.863. Additionally, melanoma cell pseudotime trajectories (average iLISI = 1.467), T cell subclustering (average iLISI = 2.204), and T cell pseudotime trajectories (average iLISI = 2.288) demonstrated well-mixed cells from different samples. The elevated iLISI scores indicate that residual batch effects were minimal and unlikely to obscure underlying biological signals. These results confirm the reliability of the downstream analyses and provide a robust foundation for investigating subtype-specific heterogeneity in melanoma.

### 4.5. Differentially Expressed Gene Identification and Enrichment Analysis

Differentially expressed genes for each cell cluster were identified using the “FindMarkers” function in the Seurat package (logfc.threshold = 0.25, adjusted *p*-value < 0.05, only.pos = FALSE, version 4.4.0). Gene Set Enrichment Analysis (GSEA) was performed using the “GSEA” function (default parameters) in the R package clusterProfiler (version 4.6.2), and Gene Set Variation Analysis (GSVA) was conducted using the “gsva” function (default parameters) in the R package GSVA [65] (version 1.46.0). The gene sets used for GSEA and GSVA were obtained from the MSigDB database [66] (https://www.gsea-msigdb.org/gsea/msigdb, accessed on 18 April 2023), including hallmark pathways, KEGG pathways, and ontology gene sets. Enrichment analysis for the Kyoto Encyclopedia of Genes and Genomes (KEGG) was performed using the “enrichKEGG” function (*p*-value-Cutoff = 0.05, q-valueCutoff = 0.05, version 4.6.2) from the R package clusterProfiler (version 4.6.2), and Gene Ontology (GO) enrichment analysis was conducted using the “enrichGO” (ont = “BP”, *p*-value-Cutoff = 0.05, q-valueCutoff = 0.05, version 4.6.2) function. The standard workflow of the R package AUCell [67] (version 1.20.2) was used to score gene set enrichment for each cell.

### 4.6. Single-Cell CNVs Analysis

The R package infercnv [68] (version 1.16.0) was used to infer CNVs in single-cell data. With parameters set to cutoff = 0.1, cluster_by_groups = TRUE, HMM = TRUE, and denoise = TRUE, we used normal melanocytes as a reference to infer CNVs in melanoma tumor cells. The CNVs score for each tumor cell was calculated by summing the copy numbers of all genes within the cell, and the degree of CNVs in tumor cells was determined based on the CNVs score of normal melanocytes. Data conversion was performed for gene gains or loss occurring on the long and short arms of chromosomes (>two copies_gain = 3, two copies_gain = 2, one copy_gain = 1, one copy_loss = 1, Complete_loss = 2), and the resulting values were summed to obtain the CNVs score.

### 4.7. Constructing Single-Cell Trajectories of Tumor Cells and T Cells

The R package monocle3 [69] (version 1.3.4) was used to construct evolutionary trajectories of melanoma cells and T cell lineage differentiation trajectories. Due to the significant batch effects present in melanoma cells, the “align_cds” function (alignment_group = “GSM”) in monocle3 was employed for batch correction. In constructing the trajectories for tumor cells and T cells, the “reduce_dimension” function (default parameters) was used for UMAP dimensionality reduction, the “learn_graph” function (learn_graph_control = list (ncenter = 1000), use_partition = TRUE) for trajectory construction, and the “order_cells” function (default parameters, version 1.3.4) for pseudotime ordering after selecting the trajectory starting point. The “find_gene_modules” function (version 1.3.4) was used to calculate gene modules with changing expression patterns along the trajectory. The “DiffusionMap” function (n_pcs = significant PCs, version 3.12.0) in the R package destiny [70] (version 3.12.0) was used to construct trajectories during T cell exhaustion.

### 4.8. Construction of Prognostic-Associated Malignant Transcriptional Regulatory Networks

The STRING database [71] (version 12.0, https://string-db.org/, accessed on 5 October 2023) was used to construct protein association networks for malignant transcriptional programs in the evolutionary trajectory of tumor cells, and k-means clustering was performed on the networks using the “Clusters” module (version 12.0). Additionally, the R package SCENIC [67] (version 1.3.1) and Python pipeline pySCENIC [67] (version 0.12.1) were used to infer transcription factors (TFs) of network nodes. The “geneFiltering” function (default parameters) in SCENIC was employed for quality control of the single-cell gene expression matrix. TFs motif rankings for the human reference genome hg38 were obtained from the RcisTarget database (https://resources.aertslab.org/cistarget/databases/homo_sapiens/hg38/refseq_r80/mc9nr/gene_based/, accessed on 10 October 2023). The “arboreto_with_multiprocessing.py,” “ctx,” and “aucell” programs and standard workflows in pySCENIC (version 0.12.1) were used to infer TFs of network nodes. The Cytoscape software [72] (version 3.10.0) was used for network visualization.

### 4.9. Drug Screening

We used the Python package Drug2cell [73] (version 0.1.2) to screen for potential drug targets and targeted drugs within the malignant transcriptional program regulatory networks. The structures of target proteins were obtained from the RCSB Protein Data Bank (PDB) [74] (https://www.rcsb.org/, accessed on 24 July 2024) and the AlphaFold Protein Structure Database (AlphaFold) [75]. For experimentally determined protein structures, we used their defined active pockets for molecular docking; for predicted protein structures, we used CASTpFold [76] (https://cfold.bme.uic.edu/castpfold/, accessed on 26 July 2024) to calculate the active pocket regions. The 3D structures of small molecule drugs were obtained from the ZINC database [77] (https://zinc.docking.org/, accessed on 28 July 2024), and protein-small molecule docking analysis was performed using LeDock [78] (version 1.0). To further annotate the screened drug candidates, we retrieved detailed pharmacological information from the DrugBank database [79] (https://go.drugbank.com/, accessed on 6 August 2024). This annotation step ensured the biological relevance and translational potential of the identified therapeutic compounds.

### 4.10. Survival Analysis

Gene set enrichment scoring for TCGA samples using the “ssgsea” algorithm in the R package GSVA (version 1.46.0). Samples were divided into high and low groups based on ssGSEA scores or gene expression using the “surv_cutpoint” function (default parameters) in the R package survminer (version 0.4.9). Subsequently, Kaplan–Meier analysis and Univariate Cox regression analysis were performed on the TCGA data using the R package Survival (version 3.5-5).

### 4.11. Cell–Cell Communication Analysis

The R package CellChat [80] (version 2.1.0) was used to infer intercellular communication, following standard procedures for data preprocessing. The “netAnalysis_signalingRole_scatter” function (version 2.1.0) was used to compare differences in incoming and outgoing communication strengths between cells. The “identifyOverExpressedGenes” function (version 2.1.0) was employed to identify overexpressed genes in cells. The “netAnalysis_signalingRole_heatmap”, “netMappingDEG” (version 2.1.0), and “subsetCommunication” functions (version 2.1.0) were used to identify signals of communication differences between cells.

### 4.12. Statistics Analysis

All statistical tests in this study were performed using R. The statistical tests used in the figures are indicated in the figure legends, and *p* < 0.05 or adjusted *p* < 0.05 is statistically significant.

## 5. Conclusions

In summary, this study provides a comprehensive comparative analysis of the molecular characteristics and genetic heterogeneity across cutaneous, acral, and uveal melanoma ecosystems. We identified key genes and mechanisms driving the malignant evolution of CM, AM, and UM, uncovered potential therapeutic drugs and targets for melanoma treatment, and elucidated the mechanisms and intercellular communication signals contributing to the dysregulation of antitumor T cell cytotoxicity in the context of immunotherapy. These findings deepen our understanding of melanoma subtypes and offer a foundational resource for future investigations into tumor biology, immune regulation, and therapeutic development across diverse melanoma contexts.

## Figures and Tables

**Figure 1 ijms-26-09956-f001:**
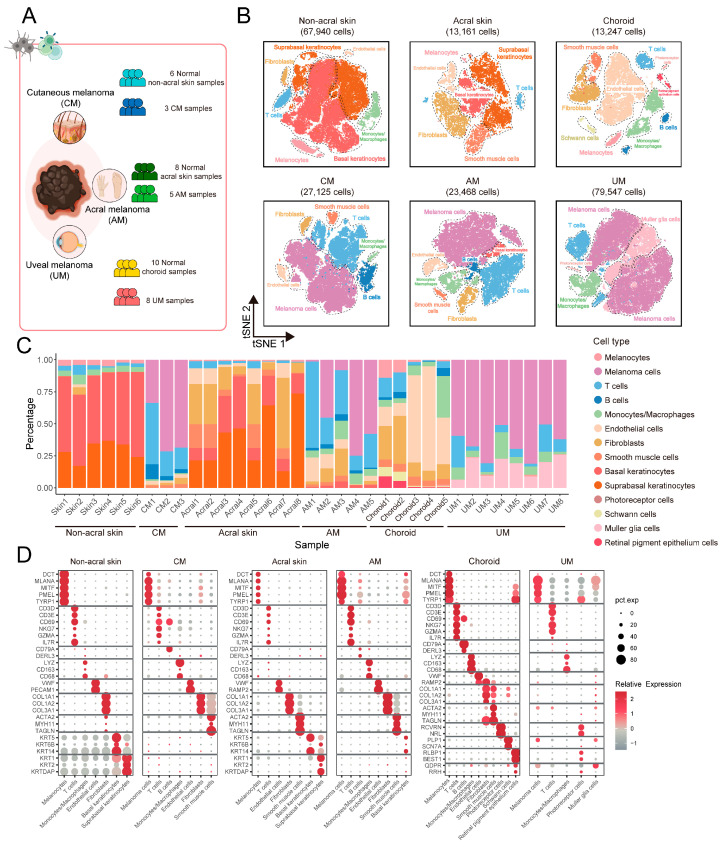
scRNA-seq profiling of the primary melanoma cellular landscape. (**A**) Overview of melanoma primary site and sample information. Single-cell RNA sequencing was applied to primary tumor site of CM, AM, and UM, alongside their respective normal tissues. (**B**) T-distributed stochastic neighbor embedding (t-SNE) projection of each site in (**A**), color codes for cell types. (**C**) Bar plot showing the proportion of cell types from samples of each site. (**D**) Dot plot showing the expression of marker genes for major cell types in each site.

**Figure 2 ijms-26-09956-f002:**
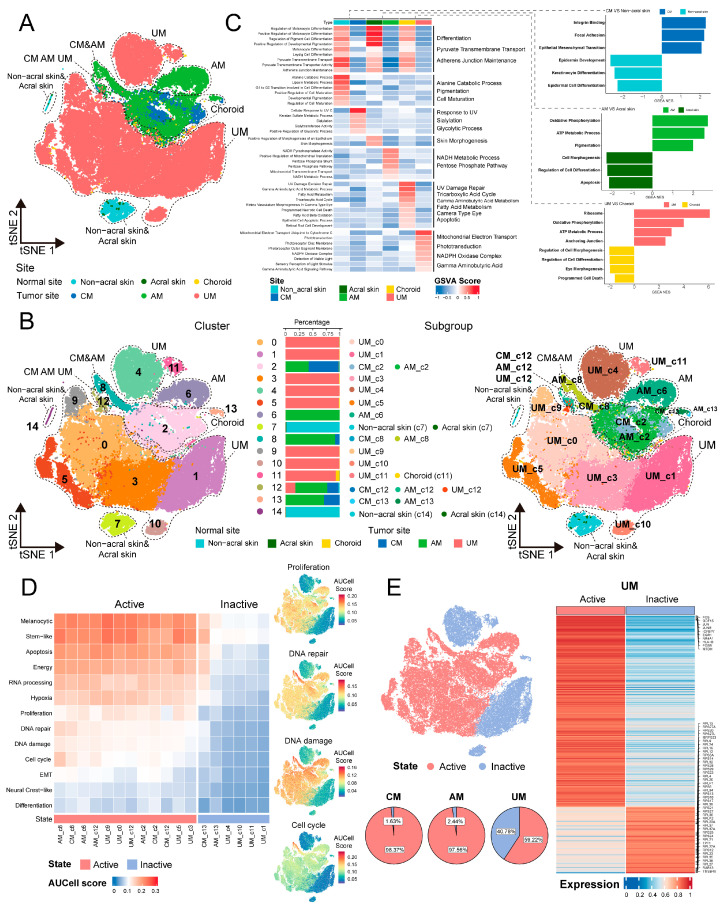
(**A**) t-SNE projection showing melanoma cell or normal melanocytes, color-coded by sites. (**B**) Subgroup division of melanoma cells based on sites and clusters. t-SNE projection showing clusters of melanoma cells and normal melanocytes (**left panel**). Stacked bar chart showing the contribution of each site to each cluster (**middle panel**). t-SNE projection showing subgroups of melanoma cells and normal melanocytes (**right panel**). (**C**) Biological function signatures of melanoma cell or normal melanocytes in each site. Heatmap showing the score of melanoma cells or normal melanocytes by GSVA in the 6 sites, including biological functions and related signal pathways (**left panel**). Bidirectional bar chart showing the significantly different signal pathways by GSEA of melanoma cells vs. normal melanocytes in each site (**right panel**). (**D**) Heatmap showing the melanoma signatures score by AUCell in the 18 melanoma cell subgroups (13 signatures) (**left panel**). Feature plots showing 4 signatures for dividing melanoma cells into active and inactive states (**right panel**). (**E**) t-SNE projection showing the cell distribution from active and inactive states, pie plots showing the proportion of active and inactive melanoma cells in each site (**left panel**). Heatmap showing the expression of significant difference gene between active and inactive tumor cells in UM, the names of the top 10 genes and ribosomal protein genes are indicated in the plots (**right panel**).

**Figure 3 ijms-26-09956-f003:**
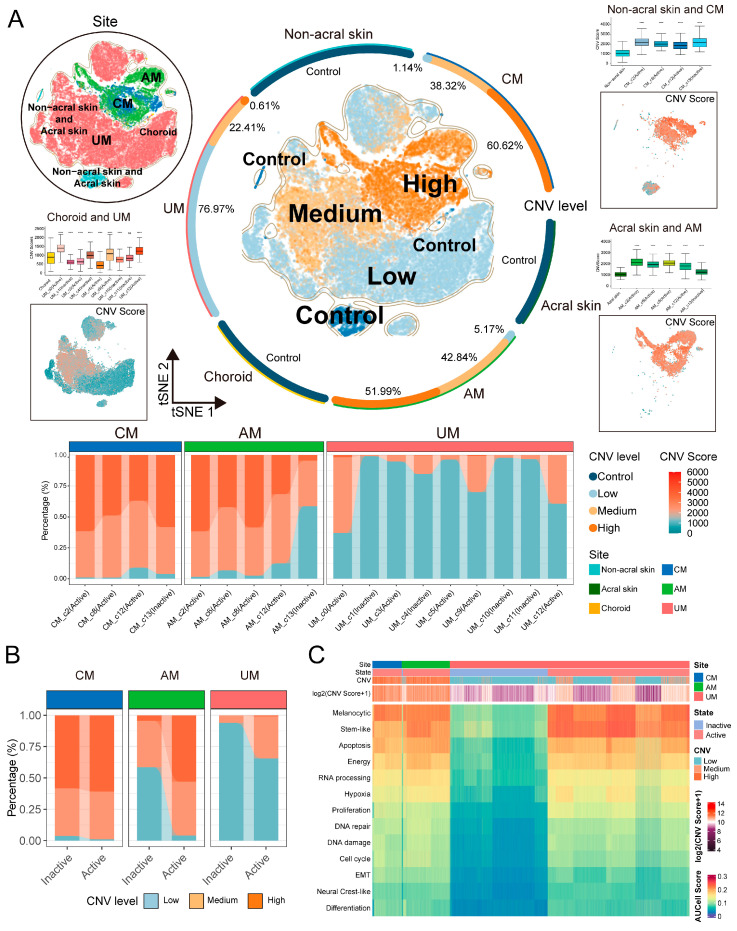
CNVs of melanoma cells in each site. (**A**) t-SNE projection showing normal melanocytes and melanoma cells, color-coded by CNVs level (middle panel, colored halo indicating the proportion of cells with different CNV levels in each site), sites (**top left panel**), and CNVs score of melanoma cell in CM (**top right panel**), AM (**bottom right panel**), and UM (**bottom left panel**), boxplots showing CNVs score of normal melanocytes and melanoma cell. Normal melanocytes as reference, two-sided Student’s *t*-test, *p*-value denoted as *, ns > 0.05, **** *p* ≤ 0.0001. Stacked bar chart showing the contribution of melanoma cells with three CNVs levels to each subgroup (**bottom panel**). (**B**) Stacked bar chart showing the contribution of melanoma cells with three CNVs levels to active and inactive states. (**C**) Heatmap showing CNVs score and AUCell score (13 melanoma signatures) of melanoma cells in each site.

**Figure 4 ijms-26-09956-f004:**
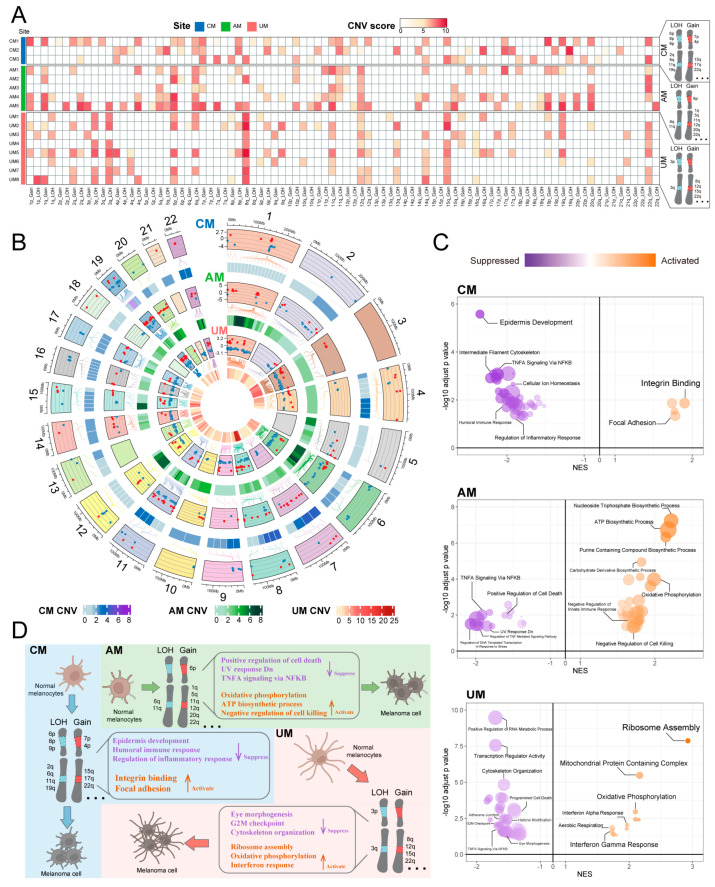
CNVs patterns of melanoma tumor cells in each melanoma site. (**A**) The summary CNVs profiles of melanoma cells for each sample inferred from inferCNV analysis (per-cell, per-gene CNVs state-level assignment). The CNVs events were categorized by gain and loss of chromosome arm. Colors in the heatmap indicate score of CNVs events in chromosome arms. (**B**) A comprehensive landscape of copy number variations (CNVs) in these subtypes, circos plot showing differentially expressed genes with CNV event (DEG with CNV) in each melanoma site. Colors indicate score of CNVs events in genes. Comparing melanoma cells and normal melanocytes, red dots represent gene upregulation (log_2_ fold change > 1) and blue dots represent gene down regulation (log_2_ fold change < −1) in melanoma cells (**top panel**) (Appendix A for all DEG with CNV). (**C**) Pathways enriched by DEG with CNV in each melanoma site. The NES and log *p* value represent significant activity of pathway enrichment, dot size represent the proportion of genes found in that pathway. (**D**) Overview of CNVs patterns of melanoma tumor cells in various sites.

**Figure 5 ijms-26-09956-f005:**
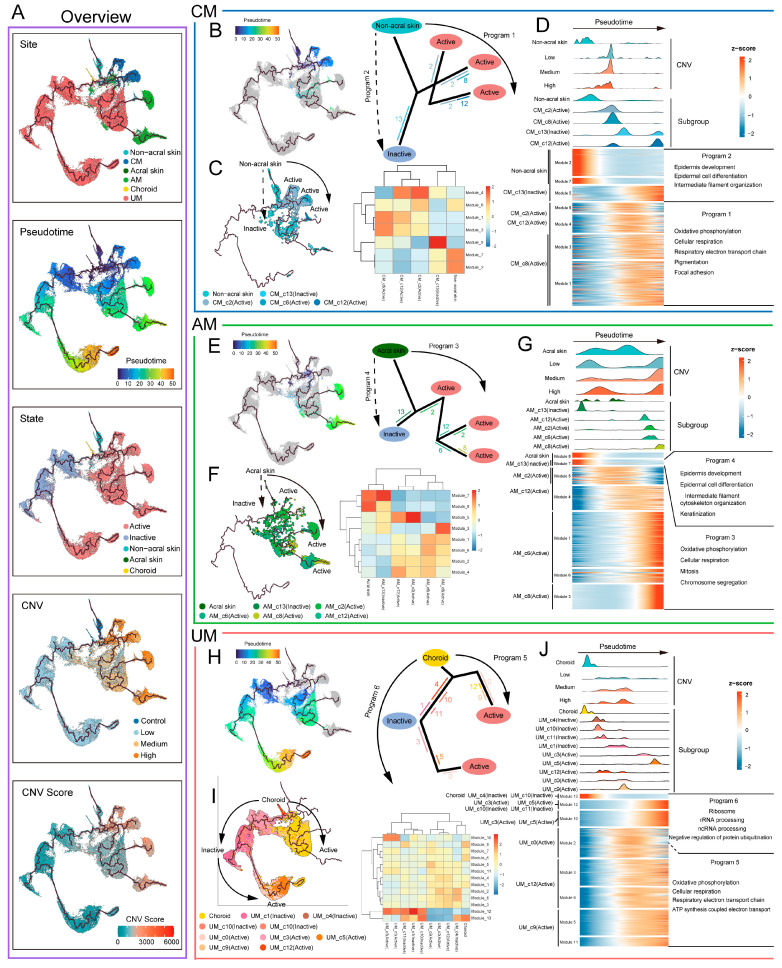
Trajectory analysis of melanoma cell in each melanoma site. (**A**) Overview of trajectory analysis of melanoma cells. Color-coded by site, pseudotime, state, CNVs, and CNVs score. (**B**,**E**,**H**) Trajectory program of CM (**B**), AM (**E**), and UM (**H**); the solid arrow indicates the potential major evolutionary direction in the trajectory, dashed arrow indicates the potential secondary evolutionary direction. The numbers and line colors indicate cell subgroups, the position of the line indicates the distribution of cell subgroups on the trajectory, and the length of the line represents the number of cells in the cell subgroup. The longer the line, the more cells it represents. (**C**,**F**,**I**) Pseudotime analysis of melanoma cells in CM (**C**), AM (**F**), and UM (**I**) trajectories, color-coded by subgroup (**left panel**). Heatmap showing gene modules of melanoma cell subgroups in CM (**C**), AM (**F**), and UM (**I**) trajectories (**right panel**). (**D**,**G**,**J**) Heatmap showing the dynamic changes in trajectory program gene expression along the pseudotime (**bottom panel**). The distribution of melanoma subgroups (**middle panel**) and CNVs levels (**top panel**) during the transition, along with the pseudotime.

**Figure 6 ijms-26-09956-f006:**
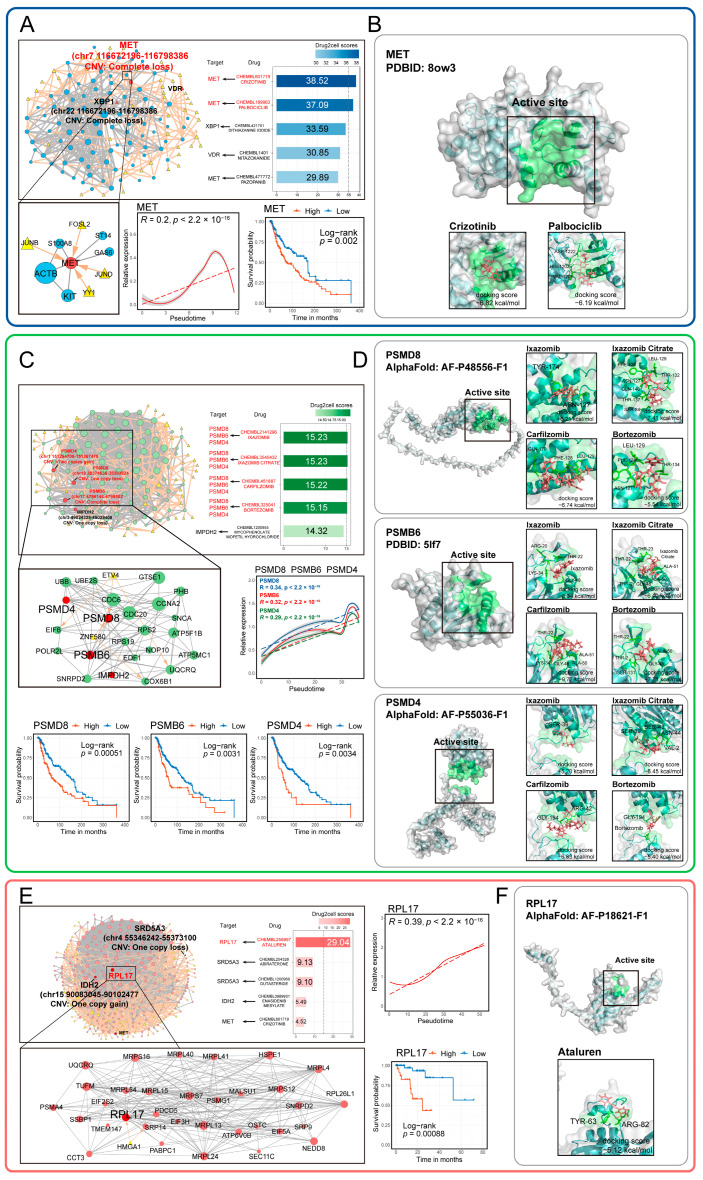
Potential drugs and targets screened based on the malignant transcriptional regulatory networks of melanoma tumor cells. (**A**,**C**,**E**) The potential drug scores, core targets, and regulatory networks identified from malignant transcriptional program regulatory networks are shown for CM (**A**), AM (**C**), and UM (**E**). The expression patterns of core targets over pseudotime are also illustrated. Kaplan–Meier analysis demonstrates the relationship between core target gene expression and overall survival in melanoma patients, with significance determined by the two-sided log-rank test. (**B**,**D**,**F**) The binding capacity and patterns of potential drugs and their corresponding targets identified from the networks are shown for CM (**B**), AM (**D**), UM (**F**).

**Figure 7 ijms-26-09956-f007:**
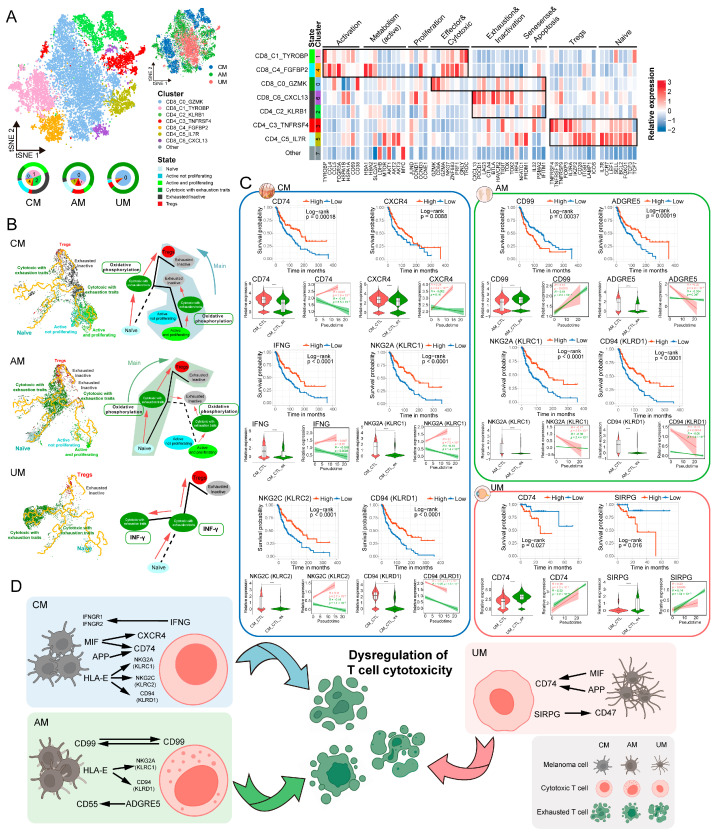
Dysregulated signaling of cytotoxicity in antitumor T cells within three melanoma ecosystems. (**A**) t-SNE projection showing T cell in three melanomas color-coded by subgroups (left panel), sites (right panel). Heatmap plot illustrating classical marker genes used for the annotation of T cell states. Pie plots showing proportions of T cell subgroups composition by melanoma site. Colored halo indicating the states of T cell subclusters and their proportions in three sites. (**B**) Trajectory of T cell in CM, AM, and UM; the solid line indicates the potential major evolutionary direction in the trajectory, dashed line indicates the potential secondary evolutionary direction. The content within the box illustrates the major signaling pathways of T cells in the “cytotoxic with exhaustion traits” state. (**C**) Ligand and receptor genes in T cells associated with melanoma patient survival, T cell cytotoxic exhaustion and pseudotime. Kaplan–Meier analysis showing the relationship between ligand/receptor gene expression and overall survival in melanoma patients, with significance determined by the two-sided log-rank test. Violin plots showing the ligand/receptor gene expression in CTL and CTL_ex. Significance was determined using a two-sided, unpaired Wilcoxon rank-sum test, *p*-value denoted as *, **** *p* <= 0.0001. Two-dimensional plots showing the dynamic expression of genes along the pseudotime, colored by CTL (red) and CTL_ex (green). The correlation is evaluated using the two-sided pearson correlation coefficient. The red and green band represents the 95% confidence interval of the regression line. (**D**) Graphical abstract illustrating the key signals of T cell cytotoxic dysfunction mediated by tumor cell communication.

## Data Availability

The raw data supporting the conclusions of this article will be made available by the authors on request.

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
