# Peer review of "Dissecting Melanoma Ecosystem Heterogeneity from Molecular Characteristics to Genetic Variation at Single-Cell Resolution"

_ijms, 2025, doi:10.3390/ijms26209956_

Round 1

Reviewer 1 Report

Comments and Suggestions for Authors

The manuscript by Hu et al. presents a comprehensive single-cell transcriptomic and integrative analysis of melanoma subtypes: cutaneous (CM), acral (AM), and uveal (UM). The authors systematically characterize CNVs, transcriptional programs, tumor–immune interactions, and intercellular communication patterns. They highlight subtype-specific CNV alterations, ribosome- and OXPHOS-associated malignant programs, and signaling pathways underlying T cell dysfunction, and further propose potential therapeutic drugs supported by molecular docking analyses. This work aims to provide a valuable resource for understanding melanoma heterogeneity and for guiding the development of targeted therapies and immunotherapy strategies.

Comments:

  • The interpretation of the ribosome–tumor suppressor axis in UM is interesting but currently speculative. Please temper the language to present this as a hypothesis rather than a confirmed mechanism.
  • The description of a “cytotoxic approaching exhaustion” transitional T cell state is intriguing but relies entirely on pseudotime inference. Please frame this more cautiously as a putative intermediate state or cytotoxic with exhaustion traits, and clarify that functional validation (e.g., cytotoxic assays, protein-level exhaustion markers) would be needed to confirm whether these cells are indeed transitional or simply represent heterogeneity within exhausted CTLs.
  • The implications for immunotherapy resistance (e.g., MIF–CD74, HLA-E–NKG2A axes) are compelling but remain correlative. The discussion should adopt a more cautious tone regarding direct therapeutic relevance.
  • Since the study integrates datasets from multiple sources, a more detailed discussion of potential biases (sample preparation, sequencing platforms, batch correction limits) would improve transparency.
  • The inference of CNVs from scRNA-seq data is informative but has inherent limitations (e.g., noise, dropouts, variability in gene coverage). Please acknowledge these constraints and discuss how they may influence the observed heterogeneity across melanoma subtypes.

Minor Comments

  • Figures: Some figures (e.g., Figure 5A, Figure 7C) have blurry and difficult-to-read legends.
  • Figure 6 in particular would benefit from reorganization to improve clarity, as the current layout is dense and may be challenging for readers to interpret.
  • While limitations are partially acknowledged, the manuscript would benefit from a more explicit section discussing the lack of experimental validation, reliance on public datasets, dataset heterogeneity, limitations of CNV inference from scRNA-seq data, and the exploratory nature of drug predictions.

Reviewer 2 Report

Comments and Suggestions for Authors

The Authors proposed a research framework through bioinformatics to analyze the tumor ecosystems of distinct types of melanoma, from molecular characteristics to genetic variations at single-cell resolution. They found oxidative phosphorylation (OXPHOS) is a critical driver of tumor cell evolution, with abnormal ribosomal gene and tumor suppressor expression observed in uveal melanoma (UM). Additionally, they screened for potential drug targets and drugs against tumor cells. In the immune microenvironment, acral melanoma (AM) and UM melanomas exhibit stronger immunosuppressive characteristics compared to cutaneous melanoma (CM). OXPHOS contributes to T cell cytotoxicity dysregulation in CM and AM, while interferon-γ is crucial in UM. Tumor cells may also induce T cell dysfunction through biological signals such as MIF-CD74 and HLA-E-NKG2A. This study offers valuable insights into melanoma heterogeneity, providing a comprehensive research framework for understanding the distinct molecular and immune characteristics of CM, AM, and UM, and potentially guiding the development of therapeutic strategies tailored to each melanoma subtype.

This study touches an important issue in melanoma biology. The single-cell analysis identified well established signaling pathways in melanoma i.e., OXPHOS and IFN-y signaling, however, the level of analysis is novel and up-to-date. Considering recent advances in the knowledge and putative targeting of these pathways in melanoma e.g., doi: 10.1186/s12943-025-02294-x, doi.org/10.1038/s41598-022-05394-6, doi: 10.1016/j.redox.2025.103552, doi.org/10.1038/s41420-025-02617-3 etc., the results and conclusions are of high novelty and clinical relevance.

The presentation of the results is very good and clearly understandable.

Discussion is up-to-date, although the Authors should consider to refer to more recent papers in the field.

Overall, the study and the manuscript is impressive and very important in the field of melanoma research.

Reviewer 3 Report

Comments and Suggestions for Authors

The research introduces a single-cell RNA sequencing comparison of melanoma subtypes. Overall, the research lacks clear novelty. The researchers compare acral, cutaneous and uveal melanoma, but this has already been done.

Introduction: The researchers fail to reference A Comparative Transcriptomic Analysis at Single-cell Resolution Reveals Acral Melanoma Features Distinct from Cutaneous Melanoma, Chinese Journal of Cancer Research and also A Comparative Transcriptomic Analysis at Single-cell Resolution Reveals Acral Melanoma Features Distinct from Cutaneous Melanoma, Chinese Journal of Cancer Research.
Patient Demographics: There is a lack of any information about the patients participating in the study. What is their age, sex, and race? Are they treatment-naïve or not?
Lack of Research Reproducibility: The code and data are both not available. This limits the usefulness of the data set to the research community and makes it less interesting. Also, there is no experimental validation of any computational analysis results. The researchers should attempt to replicate their results in the data set published in Chinese Journal of Cancer Research: "The raw single-cell RNA sequencing data reported in this study have been deposited in the Genome Sequence Archive in National Genomics Data Center of China (GSA: HRA006779)."
Lack of Algorithm Parameters Evaluation: Very little detail is given about software parameters. For example, Harmony was used to correct for batch effects. How do you know if overcorrection happened or not? In Figure 1B, the cancer cells look like one big cluster. But, that is not what should happen. Please see Figure 2a of PubMed ID 39478111. Cancer cells should cluster by patient-of-origin. Therefore, this suggests that the authors overcorrected their data and removed important biological variability by using Harmony on default parameters. Please also refer to https://x.com/simocristea/status/1931424094232179140

For these major limitations, I do not have faith in the reliability of the results.

Round 2

Reviewer 3 Report

Comments and Suggestions for Authors

The issues have been sufficiently addressed.